# Spatial Analysis of Intra-Urban Land Use Dynamics in Sub-Saharan Africa: The Case of Addis Ababa (Ethiopia)

**Amanuel Weldegebriel** [1,2,*] **, Engdawork Assefa** [2] **, Katarzyna Janusz** [1] **, Meron Tekalign** [3] **and Anton Van Rompaey** [1,*]

1    Department of Earth and Environmental Sciences, University of Leuven, Celestijnenlaan 200E-2411, 3001 Leuven, Belgium; katarzyna.janusz@kuleuven.be
2    Center for Environment and Development, College of Development Studies, Addis Ababa University, Addis Ababa P.O. Box 1176, Ethiopia; eassefat5@gmail.com
3    Center for Environmental Sciences, College of Natural and Computational Sciences, Addis Ababa University, Addis Ababa P.O. Box 1176, Ethiopia; meron.tekalign@aau.edu.et
*    Correspondence: amanueltadesse.weldegebriel@kuleuven.be (A.W.); anton.vanrompaey@kuleuven.be (A.V.R.)

**Abstract:** Currently, circa 30% of the population in sub-Saharan Africa resides in cities, and this figure is expected to double in 2040. The recent literature describes the urban expansion processes of African cities in much detail. However, the urbanization wave in Africa also leads to important intra-urban land use dynamics, which have important consequences on the quality of life within existing cities, which has received less attention. This study aims to contribute to these information gaps by (1) analyzing the extent of the urban land use conversion in contrasting urban locations using satellite images for physical criteria-based classifications and (2) assessing the potential consequences of these intra-urban conversions on the quality of life. Intra-urban land use changes were documented based on satellite imagery for the period 2002–2020. Based on some representative attributes, Addis Ababa city was selected for the case study. Urban land use dynamics and population density changes were examined based on the selected case study neighborhoods and randomly identified land parcels in the city, respectively. Urban development strategies and programs that emerged over recent decades had caused intra-urban land use dynamics, which brought significant population density changes. Moreover, these changes have caused an unbalanced distribution of socio-economic amenities across the city.

**Keywords:** intra-urban dynamics; population density; socio-economic amenities; sub-Saharan Africa; Ethiopia; Addis Ababa

## 1. Introduction

Urbanization is a prominent trend and an important issue in the world's development discourse [1]. Recent projections depict that, at present, more than half of the world population lives in urban areas, and this figure is expected to increase to 75% by 2050 [2]. The Global South countries are the hotspots of the expected urban growth. The sub-Saharan countries are the least urbanized, but have the highest growth rate, which is 4.1% per year (World Bank, 2018). The major driver of urban growth in sub-Saharan African cities is rural-to-urban migration [3]. However, many of the rural migrants searching for a better life end up in informal slum housing. The author of [4] found that in 2014, approximately 60% of the inhabitants of African cities lived in slums that lack at least one of the following amenities: (1) durable housing; (2) sufficient living area; (3) access to improved water and electricity; (4) access to improved sanitation facilities; and (5) secure tenure [5].

The high social segregation and the low quality of life in the developing world cities are considered as some of the main development gaps in the 21st century. Residential locations that are determined by socio-economic status do matter at the level of access to basic urban

amenities [6]. In a rapidly growing nation, if an urbanization process is managed with sustainable and sound planning methods, it could have paramount importance to lift millions of people out of poverty and contribute to natural resource conservation [1,7]. However, if rapid urbanization is not supported with appropriate planning policies and strong institutional setups, it may result in social, economic, and ecological stresses [8].

Most of the sub-Saharan African countries are under late urbanization, with 20–30% of their population residing in cities and towns [9]. These countries are characterized by poorly performing economies and low access to social services. However, the extent of urbanization that has occurred in recent decades is significant [10]. The urbanization drivers in sub-Saharan Africa also spring from policies employed by regimes, which could have internal or external dimensions [11] to achieve internal and external political goals. Simultaneously, urban land is commodified, therefore increasing governments' accumulation. The extensive real estate development for the high-end social class in Rwanda, the gated communities at the fringes of Addis Ababa, and the waterfront megaproject development in Luanda are a few examples [12]. Additionally, the foreign USD 2 billion debt cancelation to Rwanda and the flow of USD 20 billion per annum from the Ethiopian diaspora, of which 60% goes to real estate development [10], are some examples of foreign policies that have impacted the urbanization process in such sub-Saharan African countries.

The historical urban dynamics could be associated with specific urban development policies and strategies at different temporal dimensions. Therefore, extensive urbanization internal and external driving factors and trends have been studied from a theoretical point of view by different scholars in the discipline [13–15], while these remain insufficiently proven from empirical perspectives. Sub-Saharan Africa's urban redevelopment process encompasses urban slum demolition, resident displacement, and urban fringe farmland clearance [16]. For governments in sub-Saharan countries, where much of the income is generated from the land, slum redevelopment is considered as a good tool to change cities' image and formalize them [17–19]. The transformation of the war-torn and slum-dominated city of Luanda into a livable and attractive city [12,20], the government-led housing program in Kigali, and the slum redevelopment megaproject in Addis Ababa [10,21,22] are a few examples.

Unless urban dynamics are controlled according to the minimal sustainability standards, they could result in undesirable outcomes of urban inequality in terms of access to socio-economic amenities. Moreover, balanced access to urban socio-economic amenities is the quest for human rights within the broader framework of the Sustainable Development Goals [23]. Multi-stakeholders' involvement including urban residents in the urban changes supports achieving sustainability outcomes [24]. Although urbanization processes have been sufficiently studied in sub-Saharan cases, the internal detailed urban land use changes and the associated suburban settlement patterns and socio-economic amenities distribution are rarely covered [12,20,25,26]. Therefore, case-specific studies that depict neighborhood-level land use dynamics and the associated consequences on settlements and access to urban utilities could inform sustainable urban planning.

The diplomatic capital of Africa, Addis Ababa, which is the seat for the African Union and multinational companies, was selected as a case study area for detailed urban land use changes. Addis Ababa's representation is attributed to its similarities to other sub-Saharan African cities in its patterns of urban growth (redevelopment started from the urban core slums) and urban redevelopment agenda (legitimating political powers and foreign investment attraction in the real estate development) [10], and prominent urbanization has taken place over recent decades (after the cessation of the civil war and end of the colonial era of African countries). Moreover, urbanization statistics evidenced Addis Ababa's similar level of urbanization status to other late-urbanizing sub-Saharan countries such as Rwanda, Uganda, South Sudan, Angola, and Burundi that ranges from 20% to 30% [9].

The city, which is also recognized as the diplomatic city of Africa, has a majority population (80% of the total) that resides in slum housing, and urbanization is striding

with 8% per annum [27]. The majority of the slum residents are living within government-owned houses, and 70% of the houses are dilapidated and have poor access to basic social services and infrastructures [28]. This alarms the importance of inclusive urban development practices that ensure proper growth of the city with sufficient access to spaces and other urban amenities comparable with the population density distribution. Additionally, these practices should consider the interest of all actors, especially at the very grassroots levels [29,30].

The city's built-up expansion has been attributed to several drivers and controlling factors such as urban-to-rural migration, government-led housing development, and real estate development with the main role of the diaspora [11]. Urbanization and housing expansions are expedited by different land monetization strategies [31]. To guide the city's growth, several master plans had also been issued with successive revisions, in most cases with the support of foreign planners and architects. Such city master plans have been ill monitored according to standardized implementation methods [32]. One of the government's key urban development agendas is turning the city into a modern metropolis through urban slum neighborhood redevelopment. The majority of these slums of the city are located at the center, which was also sustained since the imperial regimes in the early 20th century [31]. The vast proportions of the houses were nationalized by the communist government after 1974 [33].

In recent decades, the government has been carrying out slum demolition in the center and farmland clearances in the periphery, where displaced people from the center are relocated [34,35]. Such infrastructural development and internal reorganizations have caused population displacements and the city's outward sprawl [34,36]. In 2005, the Ethiopian government introduced an Integrated Housing Development Program (IHDP). This entailed slum demolition in the center and low-cost housing expansion at the urban fringes [28]. The slum clearance in the center and relocation in different parts of the city have been carried out at different stages. Evidence captured on housing indicated that in the central sub-cities, residential buildups dominated by slums decreased by 15%, while housing in the surrounding sub-cities increased by 25% [22].

Inner urban core resident displacement could disconnect them from their source of employment, locational advantages, and other basic social utilities such as education, health, social networks, and livelihood bases [37,38] and cause economic risks such as food insecurity and lack of access to markets and transportation services due to a lack of properly planned resettlements [39]. Although there is supporting evidence on the urban core slum clearance and urban fringe housing expansion, information on detailed urban land use changes and the associated population dynamics and socio-economic amenities is limited. Therefore, the objective of this study was to analyze the urban land use dynamics and their effects on population density changes and residents' access to socio-economic amenities. Research questions on which parts of the city are accommodating land use changes and how these urban changes are affecting the population densities and residents' life qualities in terms of access to socio-economic amenities will be dealt with. To achieve those analyses, grid-based land use mapping and accuracy assessment [40–42], satellite image-based population density estimation [43–45], and OpenStreetMap-based urban socio-economic amenities distribution mapping were applied. The data were collected using satellite images verified through randomly sub-sampled grid cells for in situ evaluations. The essential socio-economic amenities distribution was mapped through an Overpass Turbo accessed on OpenStreetMaps. The urban land use and population density change results were compared in a spatial contrast (centers to periphery) within the temporal dimensions from 2002 to 2020. Lack of access to high-resolution satellite images that could have been used to determine the housing types, which were used to estimate population densities, was a challenge to best meet the study objectives. However, specific locations with poor-resolution images were verified through an in situ housing evaluation. The study outcomes will provide sustainable city planning insights that ensure equal access

to urban amenities and background for further urban dynamics and detailed livelihood impact studies.

## 2. The Case Study Area

Addis Ababa was identified as the capital of Ethiopia in 1886 [32,46]. It is located in East Africa (between 8°53′46.92″ N Latitude and 38°55′52.22″ E Longitude). The city has an area of 540 km$^2$, and its altitude ranges from 2000 to 2800 m.a.s.l [47]. The city is at the foot of the Entoto range (altitude 2900 m) dropping down to 2300 m in the southern periphery toward the Akaki Plains [48]. It is divided into 10 sub-cities, and 116 Woredas (districts). The land use change analysis zooms into specific neighborhoods in the urban center (Aratkilo, where the highest portion of it is located in the Arada sub-city and a small portion in the Kirkos sub-city), resident relocation area (Altad, where urban core displaced slums are relocated, and it is located in the Yeka sub-city), and the urban fringe (Yeka Abado, where the government constructs low-cost condominiums, and it is located in the Yeka sub-city). A total of 856 sample points in 36 Woredas that were exposed to land leasing, demolition, and urban periphery expansions over the last decade were selected for a population density evaluation. Based on their proximity to the center, these selected land parcels in 11 Woredas were designated as the center, and 25 Woredas were deemed to be at the periphery. The center is dominated by slums, and the periphery is a place where urban expansion takes place. The study area, case study locations, and population density estimation sample points are portrayed in Figure 1.

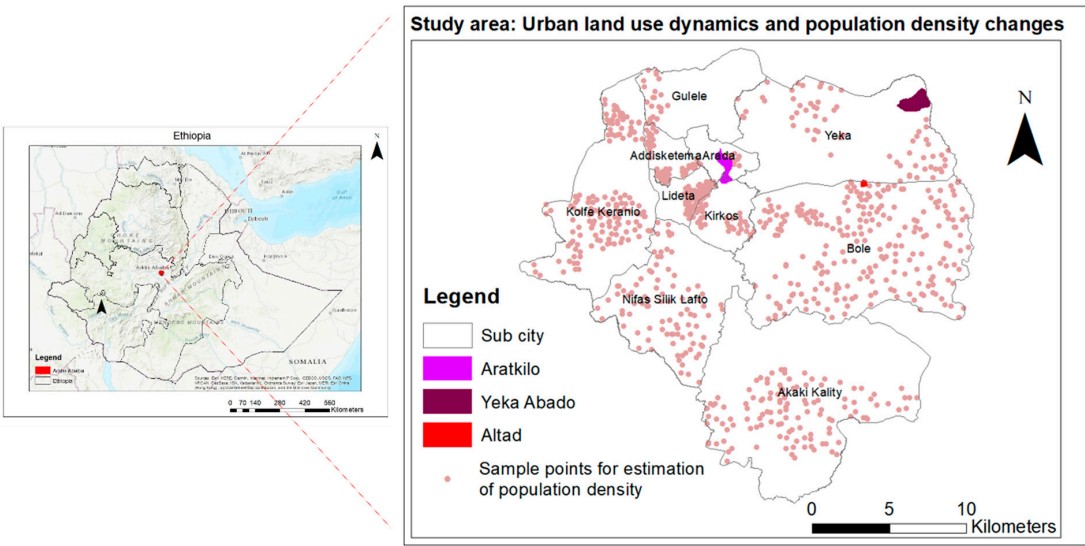

**Figure 1.** The study area, Addis Ababa, the capital of Ethiopia, is located in East Africa. The city is divided into 10 sub-cities, and the case study areas were selected from the center (Aratkilo), middle distance from the center (Altad), and the periphery (Yeka Abado).

## 3. Materials and Methods

### 3.1. Land Use Mapping

The urban land use classification in this study adopts the method applied by [40] that relies on satellite images to manually classify urban residential housing differentiations based on physical structures. The residential housing type classification in the mentioned study is based on housing features (size of the houses, number of floors and building density, quality of materials they were made out of, and other features such as streets and pavements, pools, and green areas), therefore determining the socio-economic status of the households [40,41]. Such land use classification methods were found to be practically useful as they were supported with an accuracy assessment through ground truthing. As the result of an accuracy assessment, the agreement/disagreement between the image-

based evaluated land use and the actual data is obtained by an overall accuracy value and by a kappa coefficient [40,42].

Neighborhoods that represent the city center, that is, the slum demolition site (Aratkilo = 126.92 hectare); the middle distance from the city center, which was the first slum displaced resident relocation site, which is characterized by low-rise housing (Altad = 18.27 hectare); and the periphery, which is a low-cost condominium housing expansion area (Yeka Abado = 193.75 hectare), were selected. The slum quarter is usually situated in the urban center, which is subjected to actual demolitions and is a potential target for future clearance and development into higher-quality residential and more commercial buildups. The urban fringe condominium sites are where the government and the private sector are extensively investing to expand social housing for the low and middle economic classes. We mapped land use changes in the three contrasting neighborhoods that are representative of the intra-urban dynamics. For each of these sites, historical satellite images for 2002, 2012, and 2020 were acquired and delineated on Google Earth© with 5 to 15 m resolution. These image resolution variations were observed both for different locations and different time images.

Next, for each selected neighborhood, grid cells of 50 × 50 m (0.25 hectare) were overlaid onto the delineated neighborhoods to evaluate the land use types. A grid of 0.25-hectare size was determined based on a convenience trial of different grid sizes for visibility and easy evaluation of the dominant land use features. The land uses were identified for the selected years using visual inspection and manual classification following a procedure used by the study of [40], which studied socio-economic segregations based on housing differentiations.

The three neighborhoods' land uses were identified for three time periods (2002, 2012, and 2020). (1) Slums, commercial, condominiums, bare land, green, roads, parking lots, and water, (2) low-rise houses, bare land, green, and commercial, and (3) farmland, bare land, condominium foundations, condominiums, green, traditional houses, and roads were the identified land uses for the urban center slum (Aratkilo), slum relocation area (Altad), and urban fringes (Yeka Abado), respectively (Figure 2 and Table 1). The land use maps were developed for each neighborhood at different time layers to analyze land use changes. Finally, an overlay of the compiled land use maps from 2002, 2012, and 2020 allowed us to map different change trajectories (e.g., slum to bare land, and bare land to commercial). The analysis also dealt with the conversion rates between each land use within each identified study period (2002–2012 and 2012–2020).

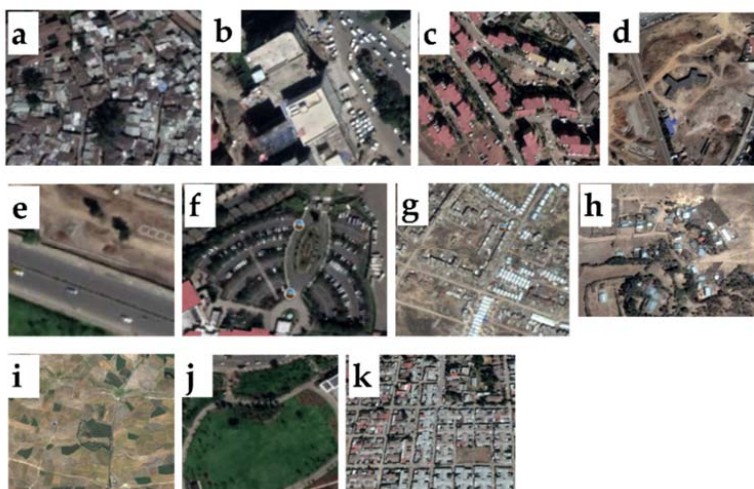

**Figure 2.** Land use classes identified for different neighborhoods. (**a**) Slums; (**b**) Commercial; (**c**) Condominiums; (**d**) Bare land; (**e**) Roads; (**f**) Parking lots; (**g**) Buildup foundations; (**h**) Traditional houses; (**i**) Farmland; (**j**) Green; (**k**) Low-rise houses.

**Table 1.** The urban land use classification criteria.

| Label | Land Use | Description |
|---|---|---|
| a | Slums | Old and dilapidated housing, high housing density, small housing size, and rusty brownish roofs. |
| b | Commercial | A building for non-specific commercial activities, not necessarily an office building. Multistory buildings for businesses, retail centers, malls, hotels, and resorts could be examples. |
| c | Condominiums | 4–6 story residential buildings, usually with external and open stairs, have nearby communal social service structures. |
| d | Bare land | Demolished areas, without buildings or any constructed site for public use. |
| e | Roads | Asphalted roads for vehicle use, in residential, commercial, and industrial areas of the city. |
| f | Parking lots | A building constructed primarily for parking cars (asphalted spaces dedicated to parking vehicles, owned by hotels or the public), which can be single-story or multi-story. |
| g | Buildup foundations | Footings of buildups under construction that never fully converted into at least single-story housing. These are neither under any residential nor business use. |
| h | Traditional houses | Small houses made out of thatched or mixed with corrugated iron roofing. Mainly situated in the peri-urban settings. |
| i | Farmland | An area of farmland used for tillage (cereals, vegetables, oil plants, flowers, etc.). These are divided into smallholding parcels for farming activities. |
| j | Green | An open green space for general recreation, which may include pitches and nets, usually municipal but possibly also private to colleges or companies. These are also spaces left for trees and shrubs to grow. |
| k | Low-rise houses | A low-rise house is a building that is only a few stories tall, and within this context, these are usually less than 4-story buildings and with a relatively high housing density. |

Source: OpenStreetMap, map features (https://wiki.openstreetmap.org/wiki/Map_features, accessed on 22 March 2021), and Wikipedia (https://en.wikipedia.org/, accessed on 22 March 2021) were used to define the identified land use features to conduct the classification. In addition, assorted land uses were in situ evaluated to identify differentiating physical attributes to verify the satellite image-based land use mapping and ensure error minimization.

The land use evaluation accuracy was confirmed based on the kappa index of agreement (KIA) [49,50], where 10% of the total grid cells were selected from the three neighborhoods for ground truthing. The accuracy assessment considered all random land use classes in 2020. Therefore, an in situ land use evaluation was carried out in the selected 10% (154 grid cells) from the three neighborhoods that comprise 97, 592, and 849 grid cells from the Altad, Aratkilo, and Yeka Abado neighborhoods, respectively. Given that intermittent ground truthing was carried out amidst an ex situ land use evaluation, the kappa of the index reached as high as 97%, which places the level of accuracy of the classification at an acceptable range.

*3.2. Population Density Assessments*

The recent national census of Ethiopia was conducted in 2007, which is not useful in providing housing density data that are closer to reality. An area-based population estimate could be used to estimate the population size when such census data are obsolete or there is limited access to demographic data [43,51]. In line with this, area-based population estimates have been used to estimate the size of hard-to-reach populations, which could be a similar circumstance with a lack of recent census data [44]. One way involves counting

the structures seen in satellite images, a method that has been tested in many settings, using both manual counting and automated counts [45].

Comparative evaluations of such area-based population size estimates against a reference population have indicated that the quality of satellite imagery determines the precision of estimates [45,52]. Therefore, housing structure counts on satellite imagery need to be supported with an in situ housing structure evaluation to minimize errors. In a location where housing structures are clearly visible, the population size could be estimated by multiplying the number of housing structures by the average individual holding size of each housing structure [44]. These methods are particularly applied by organizations working on emergency supports [44].

Therefore, to map urban population density changes with satellite images, we distributed 856 random points over 36 Woredas (the smallest administrative units) scattered across the city. Surrounding each sample point, a square grid of $100 \times 100$ m was drawn in which all the residential buildings were counted for the years 2002 and 2020. These grids were determined based on trials to evaluate the manageability of such grid sizes to count structures without redundancy or missing any housing unit, therefore avoiding potential errors.

The population densities were evaluated by counting the number of residential houses (slums, rural-type houses, condominiums, apartments, and regular housing types). Based on preliminary field observations and the reviews of the existing literature, assumptions were established to determine the household size of each house type. Within all the randomly distributed grids, the number of each housing type that was assigned a predetermined household size was counted on Google Earth images. The housing structure counts were also verified by an in situ survey for ground truthing specifically for the case of 2020. These housing types were multiplied by the corresponding household size; therefore, the population density of each grid was estimated. Additionally, the total population density in each grid was calculated and converted into per square kilometer size. Ultimately, the population density of each grid between 2002 and 2020 by spatial location was compared for significant changes based on paired $t$-test statistics.

### 3.3. Mapping Utilities and Services

Basic socio-economic urban amenities are determinant factors for the habitability of neighborhoods. Therefore, examination of the spatial concertation of socio-economic amenities has been of the used standards of urban life quality [53]. The urban socio-economic amenities distribution could affect the level of access to essential urban functions, and this is usually interlinked with the extent of the population density and quality of life [54]. Therefore, this section intends to examine to what extent the social and economic amenities are accessible against the population density differences in the city. For this case, the basic amenities were prioritized and grouped based on their importance to measure urban life qualities [55].

The list of socio-economic amenities such as financial services, health facilities and access to pharmaceuticals, access to education, and transportation services were scraped from OpenStreetMaps [56] for automatic compilation. An Overpass Turbo was used to define the queries for each identified socio-economic amenity. Each amenity was filtered by tags in the query to search all the nodes with key–value pairs specifying the amenity. The generated map features for different amenity types were converted into KML files for further processing with Arc GIS 10.7. These were imported into Arc GIS 10.7, in order to analyze their distributions against each one-kilometer distance from the urban economic center. The distribution of the identified cumulative socio-economic amenities was compared with the population density distributions along with a distance gradient from the center to the periphery.

## 4. Results

### 4.1. The Land Use Change Patterns

According to Figure 3, in 2002, the urban inner core (Aratkilo neighborhood) was dominated by slum residential houses (72%) and a smaller portion of commercial buildups including the hotel area (Sheraton Addis hotel). In 2012 and 2020, the slum showed a reduction trend to 37% and 22%, respectively. On the other hand, the commercial land increased by 9% and 19% in 2012 and 2020, respectively. The bare land increased to 40% in 2012, while it slightly decreased to 36% in 2020. Unlike the previous period (2002–2012), condominiums emerged with 5% in 2020.

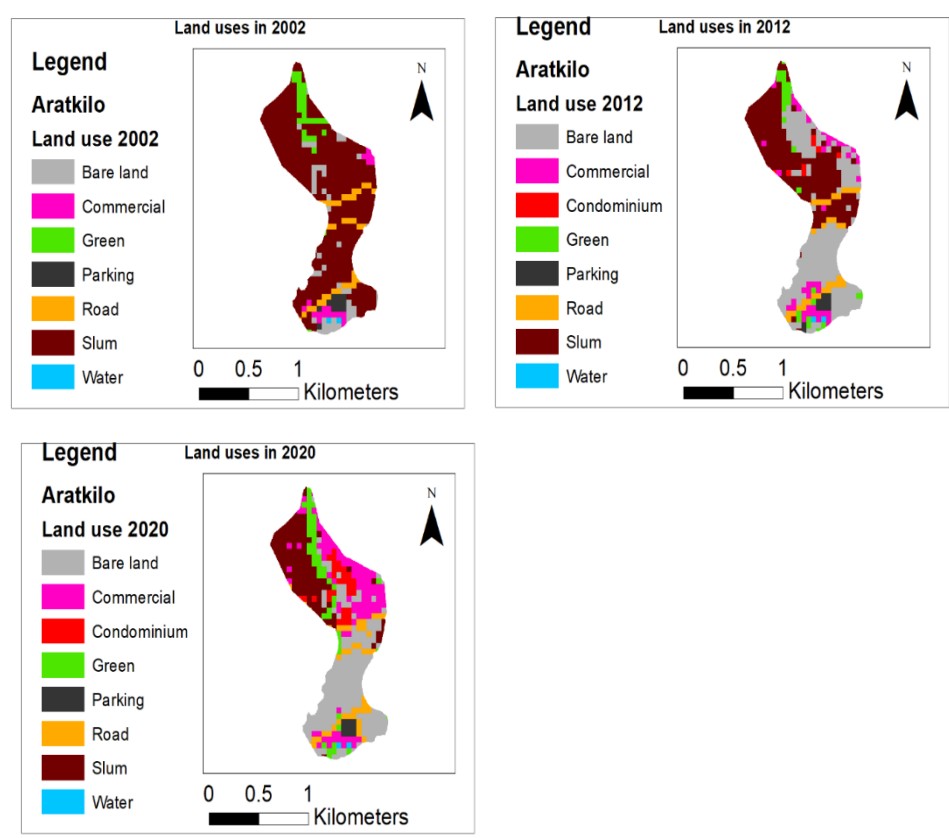

**Figure 3.** The land use composition for 2002, 2012, and 2020 in the urban core (Aratkilo neighborhood).

Slum clearance was consistent between 2002 and 2020; meanwhile, commercial buildings and government-constructed condominiums were taking over other urban land use types. Slum demolition also included some old commercial buildings that were sustained before 2002, therefore aligning with the master plan of 2016. In the neighborhood, bare land has been kept open for a decade, though after demolitions. Over the framed time period, the urban center was demolished, and the land was transferred to commercial businesses based on land leases. Due to high economic returns from the urban slum redevelopment, the urban center slums have been relocated to the urban periphery, where low-cost condominium expansions take place (Yeka Abado is one example to be presented below).

At the urban fringes, unlike in the city center, there is urban expansion mainly attributed to government-constructed condominiums and real estate development by the private sector. The condominium development was preceded by farmland clearances, land the farming community used to rely on to sustain livelihoods. The Yeka Abado neighborhood (Figure 4) is an example of urban expansion hotspots in the urban periphery. This neighborhood has relatively low land rent gap compared to the center, meaning that it is exposed to an increased population density by the low and middle economic classes.

There is a considerable count of residents displaced from the urban center slum and other registrants for such low-cost housing programs destined for the periphery of the city.

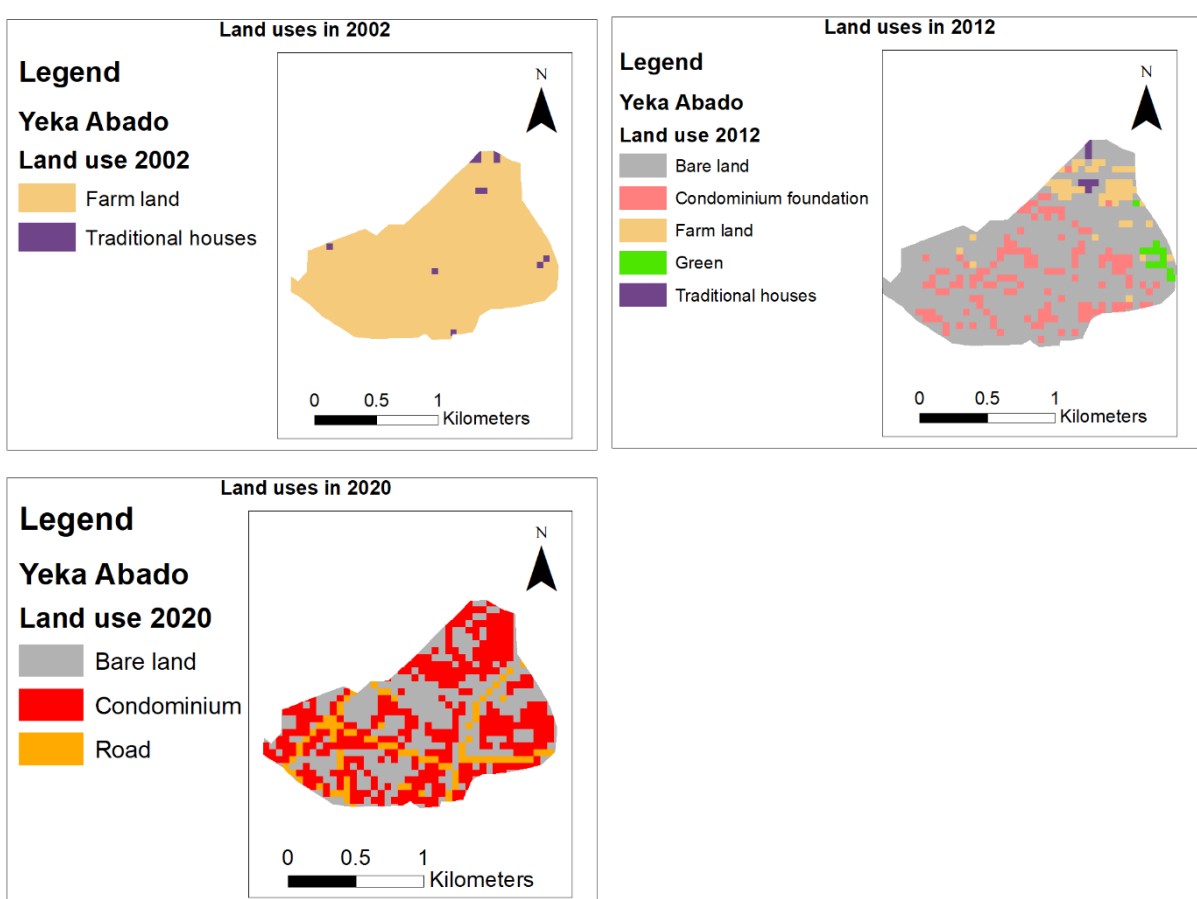

**Figure 4.** The land use composition for 2002, 2012, and 2020 at the urban fringes (Yeka Abado neighborhood).

In 2002, 99% of this area was covered by farmlands for cereal production until the government cleared it in 2012 for condominium expansion. In 2012, the area was converted into condominium foundations (16%), and the remaining land was bare land (76%) and other land types (farmland and traditional houses constructed following the condominium development). In 2020, most of the area was covered with developed condominiums (44%), open spaces (45%), and roads (10%).

The other neighborhood assessed for land use change was Altad that was used as a relocation site for the displaced urban core residents in 1997. Afterwards, the area was mainly covered with low-rise family houses (81%) and bare land (19%) that was left as a public space (Figure 5). In the land use evaluations carried out for 2012 and 2020, unlike the urban core and the urban fringe neighborhoods, there were only limited changes. During these years, there was slight housing densification from 81% to 84%, while in 2020, the roadside houses of the neighborhood (6%) were converted into commercial buildings. Conversion of residential buildings into commercial businesses is an advantage taken over basic infrastructure development and increased population growth, in order to generate incomes. Densification was also led by the government administrations through transferring open lands for small and micro-enterprises to construct sheds and provide space for small businesses. Commercialization of residential areas, including demolitions in the center, is attributed to the transition of the country's economy to urban-focused from rural-biased before 2000.

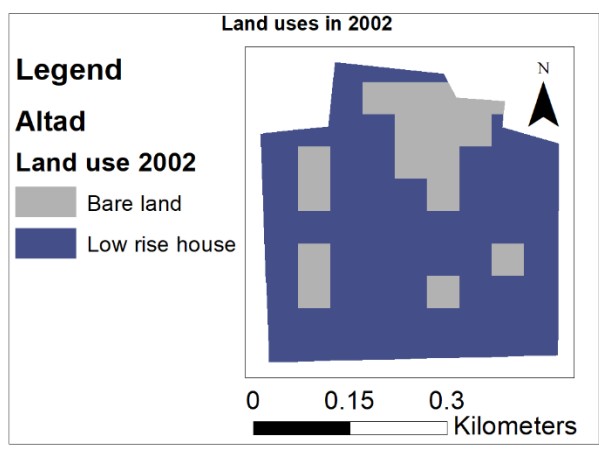

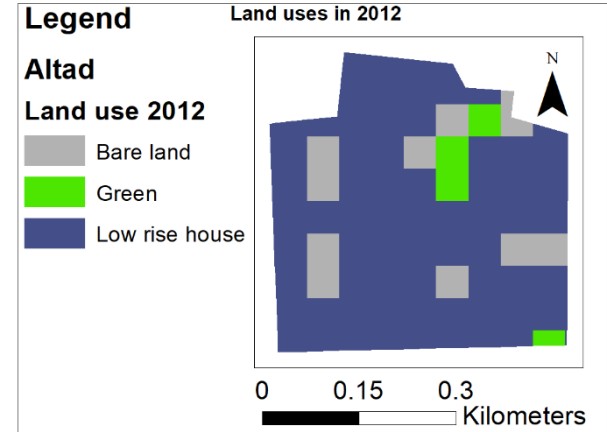

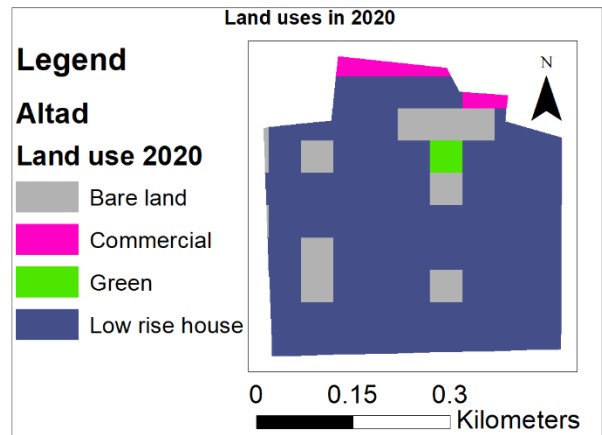

**Figure 5.** The land use composition for 2002, 2012, and 2020 in the relocation area (Altad neighborhood).

Overall, between 2002 and 2020, the land use dynamics were higher in the urban inner core and at the urban fringe condominium development sites, compared to those of the transitional zone of the city (for example, the slum displaced community resettlement area or the Altad neighborhood). In the Altad neighborhood, the land use trends seem consistent as there were no major changes from one to the other. This is what is commonly observed in the neighborhoods situated in the middle between the center and the periphery. However, at the urban fringes, there was a sharp increase in condominiums and a shrink in farmland size. The road coverage increased between 2002 and 2020 from what it was before, nil. This is due to an infrastructural expansion that could attract further settlement in the vicinity. There was a slight increase in traditional houses following the onset of condominium construction, but, finally, they were cleared by the government's actions to remove informal settlements. In the urban center, slum clearance was a continuous process from 2002. The clearance was to produce a space for more commercial and a few condominium expansions, while a substantial size of the land remained open for the last decade waiting for commercialization.

*4.2. The Spatio-Temporal Land Use Dynamics across the Two Periods*

In this section, the land use change matrix is depicted for two successive periods. Period one covers between 2002 and 2012, while period two is between 2012 and 2020. Period one coincides with the onset of the Plan for Accelerated and Sustained Development Program (PASDEP), which comprised a comprehensive urban housing program called the Integrated Housing Development Program (IHDP). Period two coincides with The Growth and Transformation Plans (GTP I and II), which have sustained the urban housing development programs. According to Figure 6, in the urban center, during period one,

48% of the slum remained untouched, while the remaining 52% was converted into mainly bare land (42%) and a few commercial (6%) land uses. Therefore, the slum, which is the dominant land use in the urban center, has been the most affected land use compared to the others. Given that the intention has been to expand commercial land uses in the urban center, the existing commercial places have remained the same to a considerable extent (86%). In the same neighborhood, but in period two (2012–2020), still a considerable amount of the slum area (57%) remained the same, except the 43% that was demolished (18%) or converted into a commercial area (12%) or condominiums (6%).

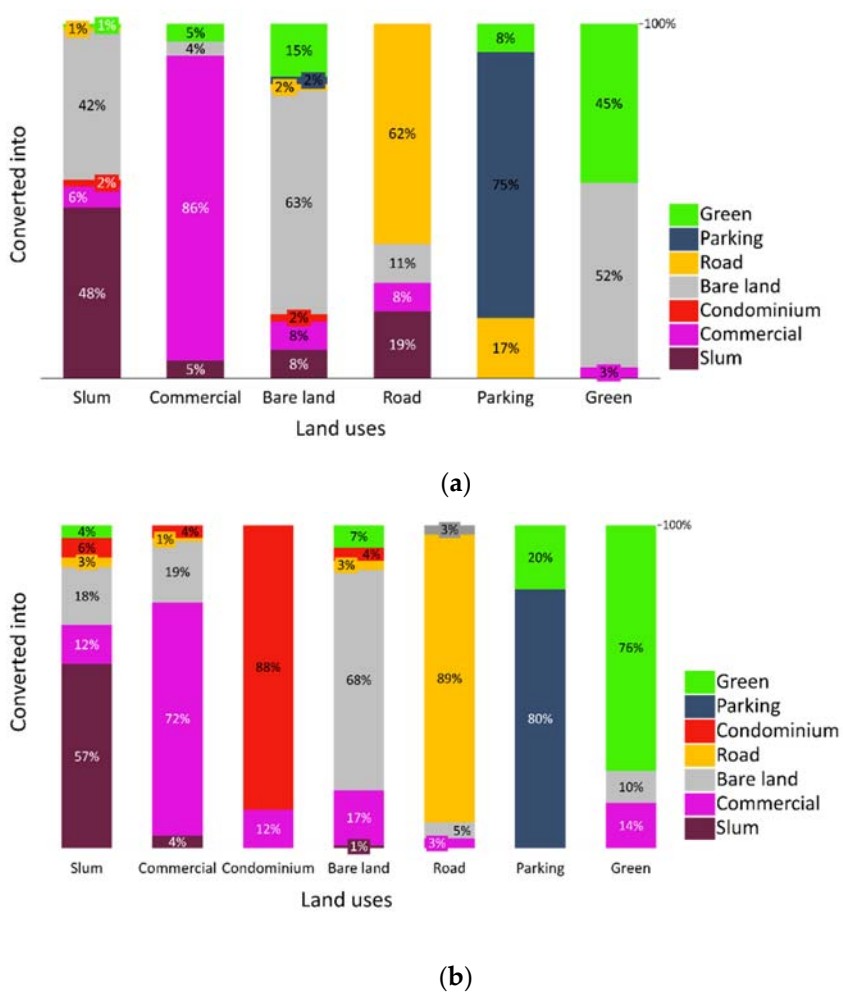

**Figure 6.** (**a**) The land use conversion matrix between 2002 and 2012 in the urban center. (**b**) The land use conversion matrix between 2012 and 2020 in the urban center.

At the urban fringe of the Yeka Abado neighborhood (Figure 7), during period one, farmland clearance (76%) and condominium housing development (17%) were the dominant changes. After 2012, the remaining farmland went into complete clearance and was converted to full-scale condominium development that entails road infrastructure. Therefore, changes at the urban fringe imply that massive urban expansion has been carried out at the expense of the urban farmland, and the low-cost condominiums are the dominant buildups.

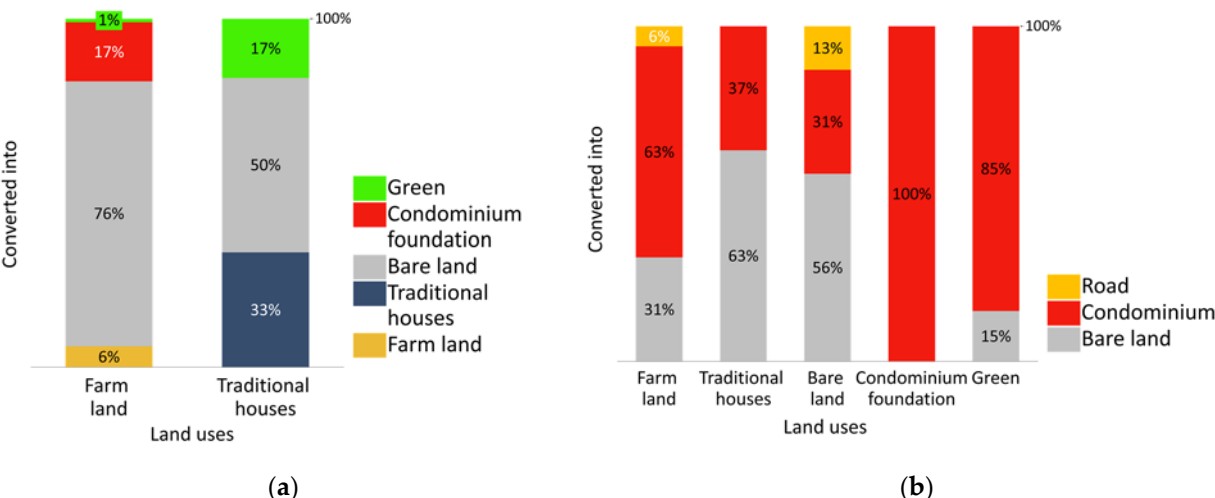

**Figure 7.** (**a**) The land use conversion matrix between 2002 and 2012 in the urban fringe neighborhood. (**b**) The land use conversion matrix between 2012 and 2020 in the urban fringe neighborhood.

The Altad neighborhood (relocation site for the urban core displaced community) underwent insignificant change during the identified periods (Figure 8). During period one, 95% of the low-rise family houses remained unchanged, while only 4% were converted into bare land. Similarly, in the following period, the low-rise houses remained with minimal changes except some residential houses facing the main road infrastructure that were converted into commercial uses and bare land in some parts, at roughly 6% each. As with some of the commercialization of the residential housing, the existing open spaces were also converted into buildups that resulted in densification in the neighborhood.

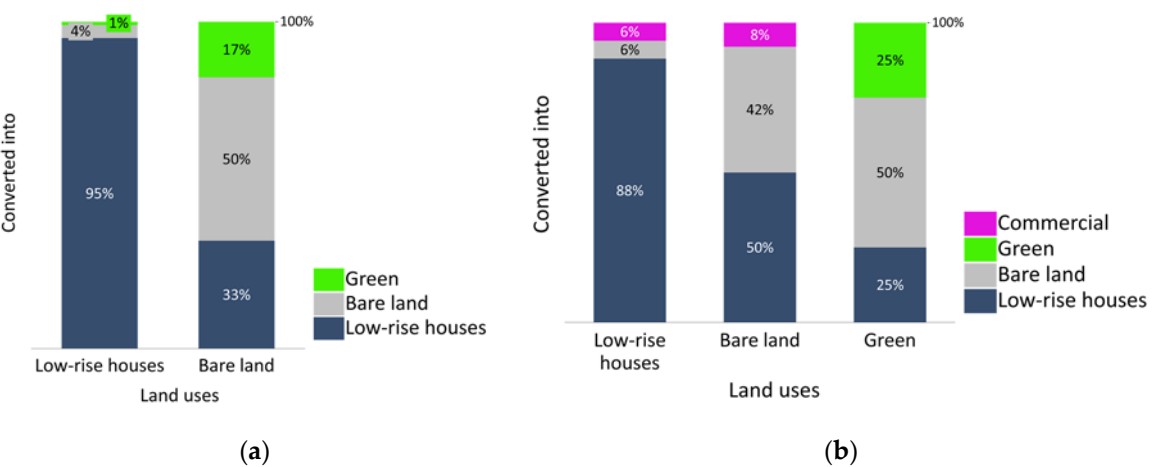

**Figure 8.** (**a**) The land use conversion matrix between 2002 and 2012 in Altad. (**b**) The land use conversion matrix between 2012 and 2020 in Altad.

Therefore, considering the changes in the study sites, the results indicate that the urban center slum and the periphery farmland were subjected to considerable changes, compared to the middle range of the city, where densification took place. The center and the periphery were much of the focus for the change because of the high economic returns in the center and the lower compensation rate of land expropriation at the periphery.

### 4.3. The Spatio-Temporal Population Density Changes

The population density evaluation was carried out based on the 856 randomly selected, but proportional to the Woreda sizes, parcels in the center and surrounding sub-cities that were significantly subjected to urban development programs (urban core redevelopment

and peripheral expansions) between 2002 and 2020. In 2020, increased urban expansion towards the periphery, coupled with slum demolition in the center for urban redevelopment, brought a relative population density loss in the center and a significant gain at the periphery. The population changes within the mentioned time dimension are depicted in Figure 9. This result is consistent with what has been indicated in the land use changes in the center and at the periphery, where slum demolition and housing expansion have taken place, respectively. These substantial urban land use and population density changes in the center and at the periphery were supported with urban development programs and policies implemented since early 2000.

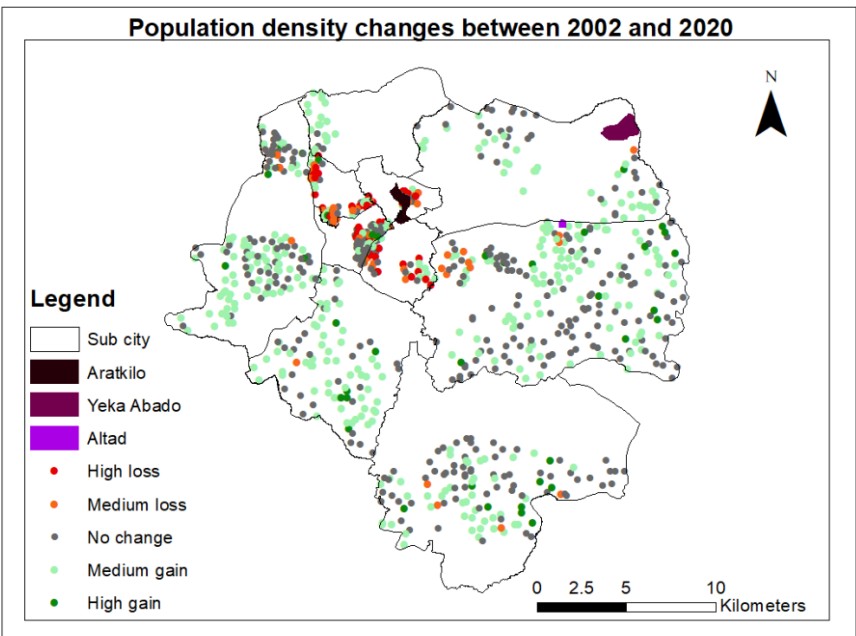

**Figure 9.** Population density changes between 2002 and 2020.

Urban development policies can not only cause land use changes but can also impact the settlement dynamics. As it is depicted, there have been slum neighborhood demolitions in the urban center, and urban expansions attributed to government megaprojects and real estate at the periphery. These results prove that there have been changes in the population density associated with slum demolition and urban social housing megaprojects at the periphery, which brought an insignificant increase in the resultant population density in the center, while a significant increase at the periphery. This is due to the broader scope of the IHDP, which also considers housing access to the city's residents in addition to the city slum residents, which are the main targets for the urban periphery housing program.

The city is divided into 10 sub-cities that fall within the central and surrounding locations based on their distances from the center. Sub-cities Arada, Addis Ketema, Kirkos, and Lideta fall under the central location, while the remaining sub-cities are mostly in the surrounding location, which is consistent with the land zonation of the city administration. This classification is not exclusive as some Woredas closer to the center are categorized within the central sub-cities while being physically located in the surrounding sub-cities. Similar to the findings in the center and surrounding locations, sub-cities in the center depicted a minimal mean population density increase, while the population density significantly increased in the surrounding sub-cities in 2020 (Figure 10). Statistical analysis (paired *t*-test) indicated that the population density between 2002 and 2020 was significant at a *p*-value of $<2.2 \times 10^{-16}$, with a 0.95 CI, where the hypothesis of an increase in the population density is accepted (Figure 10).

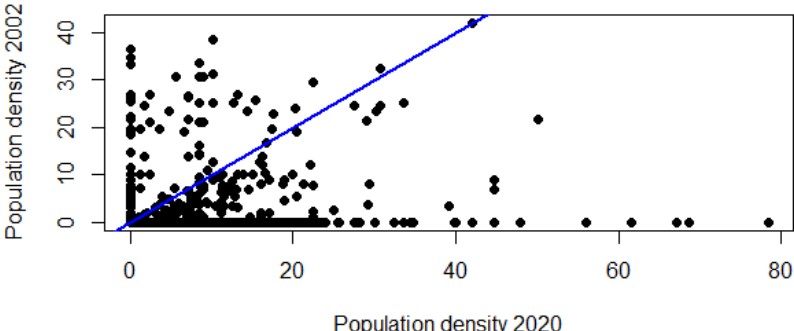

**Figure 10.** One-to-one plot of values that shows population density differences between 2002 and 2020. Circles below or to the right of the blue one-to-one line indicate observations with a higher value for population density in 2020 than for population density in 2002. Population density is in $10^3$.

The population density differences between 2002 and 2020 were also segregated into the center (demolition site) and surrounding (expansion site) locations. The population density in the center in 2020 increased from what it was in 2002, at a *p*-value = 0.09161, with a 0.95 CI, while a significant population density increase was observed in the surrounding locations, with a *p*-value of $<2.2 \times 10^{-16}$, with a 0.95 CI (Figure 11a,b).

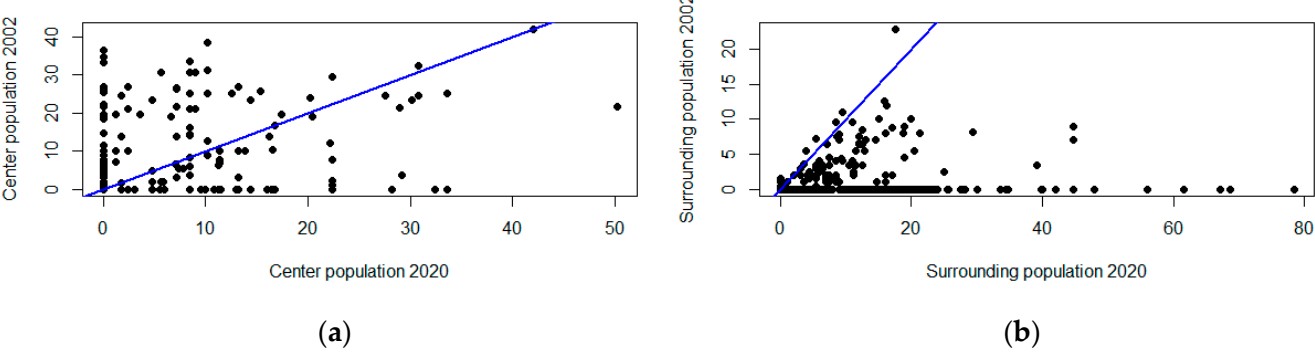

| (**a**) | (**b**) |

**Figure 11.** (**a**) One-to-one plot of values that shows population density differences in the center between 2002 and 2020. Circles below or to the right of the blue one-to-one line indicate observations with a higher value for population density in 2020 than for population density in 2002. (**b**) One-to-one plot of values that shows population density differences in the surrounding locations between 2002 and 2020. Circles below or to the right of the blue one-to-one line indicate observations with a higher value for population density in 2020 than for population density in 2002. Population density is in $10^3$.

### 4.4. Spatial Distribution of Urban Socio-Economic Amenities

Though the population density significantly increased at the urban fringes, the basic social services are still concentrated in the center, which are depicted in an amenity distribution map against distance from the center in Figure 12. The integrated housing development program, which caused slum demolition in the center and low-cost housing expansion at the periphery, yielded an unbalanced population density with the existing basic socio-economic amenities supply.

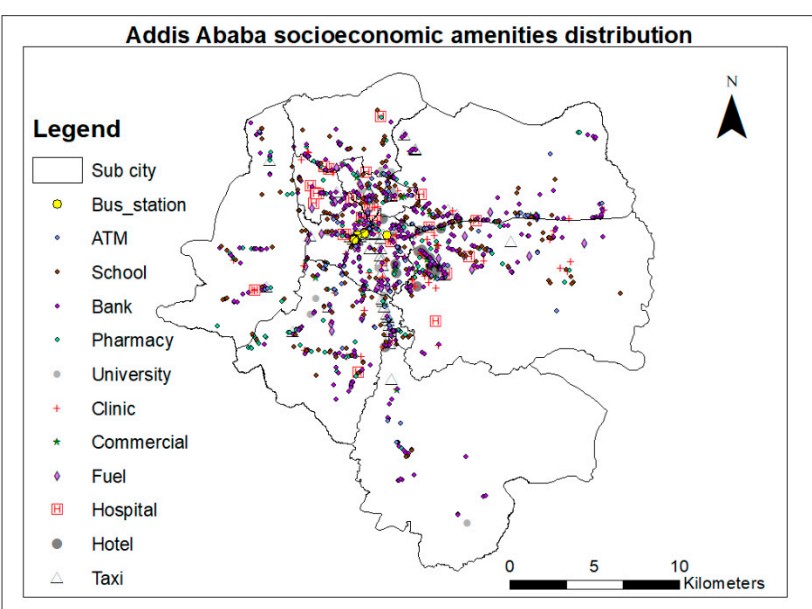

**Figure 12.** Assorted urban socio-economic amenities (health, education, transport, finance, and commercial) distribution in Addis Ababa.

The socio-economic amenities growth was exceeded by the population density, especially at the periphery. According to Figure 13 (a regression line with a 0.95 confidence band), the socio-economic amenities distribution negatively correlates with the distance from the city center. This can be attributed to the large-scale government-led low-cost housing expansions and real estate development in the urban peripheral areas. This has resulted in less socio-economic amenities access per capita at the urban periphery compared to the city center (Figure 13). This implies that the low per capita distribution of urban utilities will place the respective urban residents in a relatively challenging life situation attributed to the lack of physical access to these amenities.

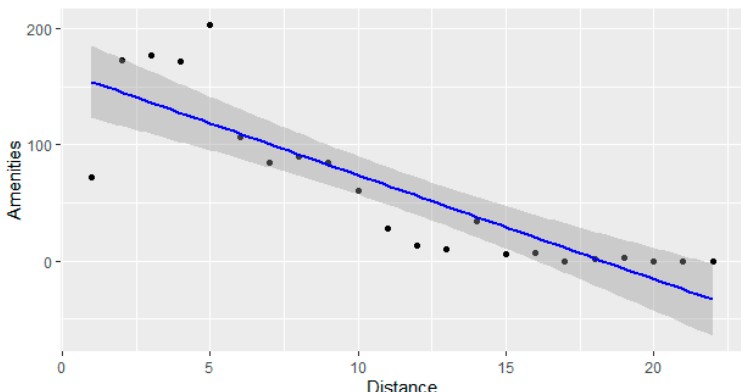

**Figure 13.** Urban amenities distribution regression (blue line) at a prediction interval of 95% against distance from the city center. Medium gray is for the prediction interval, where future observations could fall, and black circles are for the total count of assorted amenities at every one-kilometer distance from the center.

## 5. Discussion

Sub-Saharan African cities' recent trends of urbanization that are focused on the redevelopment of slum neighborhoods with an aim of city image change and legitimating governments' political power through an urban development-based economy were found to be the key drivers of slum demolitions in the urban centers [57]. In addition, suburban

expansion areas are targeted to generate financial revenues through land monetization from private real estate development and lower-class and middle-class housing programs. Studies on urban inequalities indicate that urban redevelopment-led programs push the urban poor to the urban peripheries, where urban amenities are yet to develop [58], which is practically in line with the findings of this study. Urban development policies and strategies that have been followed by the sub-Saharan African cities have shaped the urban dynamics, especially at the urban inner core, for commercialization and considerable peripheral expansions with limited access to amenities.

To deal with urban changes and their socio-economic impacts, the expensiveness of high-resolution satellite images and the unavailability of census data, especially in the resource-poor sub-Saharan African countries, have hampered research that could inform pragmatic urban planning. This study has revealed that through the application of low-cost and remotely workable methodologies to document urban land use dynamics, their consequences on population density changes and access to urban socio-economic amenities are possible to estimate through proxy variables. Other studies have also proved that similar methodologies could be valid and useful approaches to deal with urban land use mapping [40,41] and population density estimations [43–45,52]. Additionally, the limitation on the urban land classification could be the lack of very high resolution satellite images such as IKONOS and Quickbird imagery up to 1 m resolution, which could be the best fitting images for urban dynamics analysis based on object-based urban analysis that uses both spectral and spatial information for classification. Having varied resolutions for different locations was also another limitation faced when working with Google Earth images [59].

Moreover, the distribution of urban socio-economic amenities and spatial variations were also mapped by [44] based on the existing administration data in urban amenity records, whereas in this study, we employed an innovative method to scrape urban socio-economic amenities georeferenced spatial data from open data sources (OpenStreetMap). Therefore, this renders inferences more realistic and minimizes errors as opposed to the socio-economic amenities data that might be documented in administrative localities.

In line with the findings of this study, in sub-Saharan cities, the closest settlements to the center and urban encroaching areas are riskier for change with negative implications when landholding policies favor the state [22,60]. On the other hand, peri-urban residents in cities, where there is private land proprietorship, take advantage of outwards urban expansions through land sale and engagement of productive economic activities [61]. Comparatively, the increased population density at the periphery than in the center is also supported by findings from other studies, which justified that urban farmland and open spaces are easier to turn into buildups as the farmland compensation for expropriation is lower [62]. This could have also caused a higher rate of population density change at the periphery than in the urban center, where resident displacement holds higher government expenditures. The urban renovation processes in the urban inner core slums and social housing expansion at urban fringes could have merits in terms of the city's image change and affordable housing for low- and middle-income residents; however, unless such urbanization is supported by strong institutions grounded on the premises of the constitutional land use rights, and responsible institutional structures with the required capacity, it will have negative social and economic outcomes [16].

Minimal public and other stakeholder consultations, unbalanced interest between the government priorities, and the overlooked resident community [63] were also documented to be some of the causes of the negative outcomes of urban dynamics in sub-Saharan African cities. Cities are the centers of national economies; therefore, their governments' economic development is highly tailored to urban land [64]. This is a clear indication of the government maintaining its financial interest in urban land and housing development. Additionally, in similar and validating studies in Kenya, the National Housing Policy, which promoted slum upgrading with minimal displacement of people, though, eventually, the goals were never met, has changed the population dynamics [65]. The author

of [66] identified that in some sub-Saharan countries, public housing development is a leading factor to determine the cities' shapes, and the author recommended the issue that sustainability needs to be the pillar for urban development.

Access to urban social and economic amenities such as financial and market services, water and electricity supply, and sufficient transportation [55] is directly interlinked with human life qualities and sustainable development indicators [44,67,68]. However, the existing amenities distribution in the city indicated that residents situated at the periphery have comparably limited physical access to these urban amenities. This finding is agreeable with the theory of uneven development that has been explained and evidenced by studies in [69]. This theory identifies uneven development at different spatial contrasts as a driver for an unequal distribution of socio-economic amenities. Such uneven development and unequal spatial distribution of urban amenities could be some of the ways to depict urban inequalities, which is an undesirable outcome of sustainable development caused by urban changes [70–72]. Therefore, this could be counted as vivid evidence for socio-spatial inequality that leaves the urban sustainable agenda unattained, and it could be addressed through good urban governance and efficient institutions that could align urbanization with sufficient socio-economic amenities access [73,74]. Detailed livelihood impacts of urban land use changes could be a follow-up study to strengthen the evidence for the need for sustainable urban planning.

## 6. Conclusions

The literature and evidence from practical cases indicate that the slum sections of sub-Saharan African cities are, at present, the economic priorities for redevelopment. Such slums are closer to the economic center, where high economic rent could be accumulated by government housing programs. On the contrary, the late-urbanizing sub-Saharan African cities run housing megaprojects that take place in the urban expansion areas. These cities' peripheries are dominated by low-cost social housing inhabited by people from low economic classes including those displaced from the urban center. This has resulted in an increased population density over a lower intensity of urban socio-economic amenities compared to the center, which might be a contributing factor to socio-spatial inequality unless curbed through sustainable urban management practices.

The urban inner core slum and the urban fringes of Addis Ababa are the city portions where higher government-led urban land use dynamics have occurred. The inner slum near the political center of the city has been the priority target for redevelopment over recent decades. However, Google Earth imagery shows that much of the area has been used for commercial reasons and partly kept open for future land leasing and accumulation. The demolished inner core urban slum is mostly designated for a mixed residential facility; however, larger proportions are being used for retail and commercial buildings due to its high land rent potential compared to the residential purposes. The slum residents are displaced offsite to the periphery low-cost housing. Therefore, this has already created a high population density in the suburban residential areas, while the central commercial and business areas remain unused during off-work hours.

Urban development programs and strategies in such sub-Saharan African cities that are running to redevelop their cities need to consider relocations with minimum negative impacts on the displaced community and even, if possible, redevelopment without displacement, through intensive high-rise residential apartments, whose rent value could be shared and affordable through some government subsidization strategies [75]. This could help urban centers to absorb higher population densities and relieve population pressure in the periphery. Additionally, unplanned urban sprawls, where the city's expansion is exceeding the sufficient distribution of urban amenities, need to be controlled to ensure residents' sustainable access to urban amenities. Moreover, the institutional structures at different levels within the municipality have to be reframed to realize expansions that are supported with the required urban facilities and controlled over urbanization that could put pressure on the equitable access to urban basic amenities. Additionally, urban

redevelopment planners need to pay sufficient attention to ensure institutional setups to accommodate effective feedback exchange systems during the entire urban dynamics, including community and relevant stakeholder participation, regular monitoring, and evaluation against sustainable urban development standards, are in place [76]. In addition, informed and effective institutional setups that support sustainable urban redevelopment could benefit from further in-depth research on slum clearance and its impacts on the local livelihoods.

**Author Contributions:** Conceptualization, A.W., K.J. and A.V.R.; data curation, A.W. and M.T.; formal analysis, A.W.; methodology, A.W. and A.V.R.; project administration, A.V.R. and E.A.; resources, A.W., E.A., K.J., M.T. and A.V.R.; supervision, E.A. and A.V.R.; validation, A.W., A.V.R., K.J., M.T. and E.A.; visualization, A.W.; writing—original draft, A.W.; writing—review and editing, A.W., E.A., K.J., M.T. and A.V.R. All authors have read and agreed to the published version of the manuscript.

**Funding:** This research was funded by VLIR UOS (https://www.vliruos.be/en/home/1, accessed on 15 July 2021) through the Global Minds Scholarship of KU Leuven University (https://www.kuleuven. be/global/global-development/funding-possibilities/globalminds, accessed on 15 July 2021).

**Institutional Review Board Statement:** Not applicable.

**Informed Consent Statement:** Not applicable.

**Data Availability Statement:** The data presented in this study are available on request from the corresponding authors.

**Acknowledgments:** We are grateful to the Global Minds (the VLIR UOS)-funded program, from the Belgian government, through the cooperation of the University of Leuven. We would like to express our deep gratitude to Addis Ababa University, College of Development Studies (the local institute), which supported the research process.

**Conflicts of Interest:** The authors declare no conflict of interest. The funders had no role in the design of the study; in the collection, analyses, or interpretation of data; in the writing of the manuscript, or in the decision to publish the results.

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
