# Peer review of "Spatial Analysis of Intra-Urban Land Use Dynamics in Sub-Saharan Africa: The Case of Addis Ababa (Ethiopia)"

_urbansci, doi:10.3390/urbansci5030057_

Round 1

Reviewer 1 Report

Major comments

The subject matter of the paper is highly relevant to topics including slum mapping, urban land-use changes. The deep investigation of urban neighborhoods in Addis Ababa also interesting for the community. However, I have many concerns relating to:

  • The background of the research was set up in a Sub-Sahara Africa context when the research is focused on Addis Ababa neighborhoods. It is not clear how the local study is representative of the region
  • A lack of clarity around the research questions
  • Inadequate literature review on existing spatial analysis or mapping methods
  • Method on classification was not elaborated and the method on population density assessments seems very empirical  
  • The result section presents many figures but lacks further analysis
  • The discussion section read like background information, not closely related to the results of this study
  • Many texts in sections 2 and 5 would fit better in the introduction

These concerns would need to be addressed by major revisions to all sections. Detail is provided below.

1 Introduction

  • Context information on global and Asia is not so much related to this study.
  • This Section presented background information on Sub-Saharan countries, later section 2 presented another background information on Addis-Ababa. It needs to explain how these two sections are related, or consider merge these two sections.
  • This section reported many problems in the urbanization process; however, it is not clear which exact question your study is addressing. Setting up your research questions would be helpful.
  • There is an abrupt switch from background information to research goals. It looks like the research focuses on mapping slum and related land-use change, however, literature on similar or existing mapping methods is missing.

2 The case of Addis-Ababa

  • This section directly go for information on Addis-Ababa without explaining why it is chosen as a case study city.
  • The background information on the city would be more suited in the introduction section unless it is related to your method.
  • Figure 1 needs to be improved, the text on the figure is barely legible.

3 Materials and Methods

  • The focus neighborhood Aratklo and Altad are not presented on the map of the study area (figure1)
  • The classification procedure as an important part of the method was not elaborate at all.
  • The method of estimation of the population is every empirical, any reference could support its applicability?
  • It seems that the mapping procedure involves many manual works. Any measures to reduce or control errors in the process?

4 Results

  • This section describes the many changes in figures for each neighborhood. However, further analysis or comparison across neighborhoods would help the readers to learn from your findings. Some of the figures are better suited as supplement material.
  • Some references or quality evaluation methods would give readers an impression of the accuracy of your results.

5 Discussion

This section goes to general background information again. It might be more appropriate to refer discussion to your methods and results, base on the limitations, practicalities and potential of your research.

6 Conclusion

The conclusion started with the city of Addis Ababa, are the suggestions afterward also given to the city only?

The conclusion could be closer to your study results, make it clear what are the main findings, contribution to existing studies, and the outlook for the future.

Reviewer 2 Report

The paper analyzes the variation of land use composition (since 2002 to 2020) in the urban core of Addis Ababa (Ethiopia).

The following revisions and suggestions are recommended:

  • Figure 6 (a and b). The legend is not clear. Avoid to write: comm ercial, condo  menium, etc., but insert a word per line.
  • Figure 9 is not perfectly readable because the resolution is too low. It is not possible to distinguish the different colored points in the legend and in the map.
  • Figure 10 is not clear. Explain what the abscissa means: what is factor (locations)? And also what the ordinates are? And the blue area? And the black points? Avoid using “tot-pop-dens_2022” but name exactly the unit of measurement. Distance in km?
  • Figures 11 and 12 are not clear too. See previous comments about Figure 10 and explain, in this case, what “index” is.
  • Figure 12 (as Figure 9) is not perfectly readable because the resolution is too low. It is not possible to distinguish the different colored points in the legend and in the map.
  • Figure 14. Explain it better. Distance in km? Avoid “All_amenities” (see previous comments)
  • Check editing and typos.

Reviewer 3 Report

Authors present a very interesting research on the analysis of "intra-urban land use dynamics" considering the time period 2002-2020 in Addis Ababa city. In my opinion, the main theme, the mapping and the description and analysis of land use modifications in 3 different neighborhoods (urban center, urban fringe and relocation area) represent a fundamental starting point to understand the dynamics of urban development (from top-down and bottom-up perspective and from public/private point of view, considering social, cultural and economic aspects) and to forecast (and/or define) the future trends especially in developing countries.

The paper is well structure and clear in each sections. Nevertheless I suggest a language, spelling and punctuation check from a native speaking.

Main issue: authors state that this research wants to assess "the potential consequences of these intra-urban conversion on the quality of life". In my opinion this goal is not completely satisfied because it is not well remarked how the land use changes or the population density variations affect the quality of life in each neighborhood. Authors describe some social/economic aspects in the discussion section but it has to be improved.

Minor issues:

  • in the final part of introduction section, explain better method and paper structure to facilitate the readers' understanding;
  • the caption of Figure 1 is not complete, please revise;
  • in paragraph 3.1, it can be helpful insert the dimensions (square kilometers or hectare) of each area;
  • as explain in paragraph 3.1, authors use a grid cell 50mx50m. Why do authors chose this grid? Availability of satellite images? Any images resolution problems? Morphological-typological characteristics in each area? Please explain;
  • In relation to paragraph 3.2, are not available census data for the population density? (i.e. number of residents or inhabitants per square meters of gross floor area). Moreover, do authors consider the "number of each housing types" (line 200) only from satellite or Google images or do they make in situ surveys (for 2020 case)?
  • Figure 3a: pay attention to colors similarity between legend and chart (slum and parking);
  • Figure 6a: about green area chart, is it correct that two different percentage have the same color? Please check.

Round 2

Reviewer 1 Report

The previous overview stands. Some concerns remain or emerge in the revised paper such as:

  • The introduction section could be better structured, intent and research questions are not clear.
  • Methods related to remote sensing classification seems problematic
  • The result section could be more informative

These concerns and major comments that are set out below would need to be addressed by revisions before I would be confident to recommend the paper for publication.

Major comments

Abstract

  • “innovative remote sensing methods” is not convincing, see the comments for the methods section

Introduction

  • line 54 to 66 foreign investment is not so much relevant
  • line 79, already in the 5th paragraph, it is still not clear what the paper is about or what the research sets out to do.
  • line 92-134, the information on Addis Ababa urbanization could be revisited. If Addis Ababa is representing sub-Saharan African cities, the background on sub-Saharan African urbanization has been reported, no need to be repeated for case study city.
  • The research questions are not related to the background information. What is the goal of this study, what are the challenges to realize your goal?
  • line 153-162 seems not to belong to the introduction section

The case study area

  • Figure 1 legend “population density” should be the sample point for estimation of population density

Materials and Methods

  • line 194-202 this section starts with summarizing and criticizing three references, without explaining how it is related to your methods.
  • Line 217, “The land uses were …classification following a procedure used by [40].” Reference 40 used Landsat as input data and maximum likelihood supervised classification. If your study follows the same procedure, please clarify your input data and the classifier.
  • Since you have done land use maps for three different years, the accuracy assessment was carried out for which year and which land use classes are involved in the accuracy assessment?

Result

  • Line 345-352, “in 2012 the slum has been reduced to 37%...in 2020, the slum continues to reduce into 22%...” It would be better to put these two parts together, as the trend of change is the same. The same for other land use classes.
  • The left and right pictures in Figure 3a are not necessary. The same for Figures 3 b and c. After reduction, figure a, b and c could be combined as one.
  • Section 4.2 should be better structured and more compacted. The change from and to slum seems most important thus should be the focus, the description of other classes could be shortened.
  • Line 505-514, during periods one and two, if the trend of change remains the same, the description for some land use classes could be summarized, to make the text more informative
  • I believe figure 6 a, b and c are not the best way to show land-use changes. These figures can not illustrate the trend of the change, either a comparison between different land-use classes.

Discussion

  • Many parts of the discussion repeat the information from the result section, whereas it should contain points of interest for discussion or reflection. It would be easier for readers if you set up the three or four points and then re-structure this section accordingly.
  • Line 720 “time effective” is not convincing, as many manual works were involved.
  • Line 724, “lack of high resolution satellite images such as IKONOS … for classification”, it was never mentioned what is the spatial resolution of the satellite images used in your study, so it is hard to judge whether IKONOS will improve your results.

Author Response

Comments from Reviewer 1

Dear reviewer: Thanks again for the review and pointed out comments for the second round. The responses to your comments are addressed point by point (with blue highlights). Each comment is followed by its respective response. The line number or sections are also indicated for you to trace changes made in the manuscript. Your general comments are combined with the detailed comments, but they are also underlined. In addition to the comments below, grammatical and spelling corrections were

Sincerely,

The previous overview stands. Some concerns remain or emerge in the revised paper such as:

These concerns and major comments that are set out below would need to be addressed by revisions before I would be confident to recommend the paper for publication.

Major comments

Abstract

Comment 1

  • “innovative remote sensing methods” is not convincing, see the comments for the methods section

Response 1

  • Yes we agree: the innovative aspects of this paper are not the technical remote sensing methods but their applications in an urban development context. Therefore, it is omitted and restated as “satellite images for criteria based classifications” instead of “innovative remote sensing method”.

Introduction

  • The introduction section could be better structured, intent and research questions are not clear.

Comment 2

  • line 54 to 66 foreign investment is not so much relevant

Response 2

  • We agree and  shortened this section as follows: “urban land is often commodified to increase government’s accumulation” Line 60

Comment 3

  • line 79, already in the 5thparagraph, it is still not clear what the paper is about or what the research sets out to do.

Response 3

  • As an urban dynamics could put an impact on socio-spatial structures, human centered urban developments should consider the right of residents to adequate housings and associated socioeconomic amenities, therefore there has to be a balanced urban land transformations in line with international human rights [23]. In the sub Saharan Africa, slum upgrading process involves the government, residents’ representatives and different actors that are offered a contract for different scope of work, therefore efficient urban institutions are required [24].
  • The above paragraph changed into
  • Unless urban dynamics is controlled according to minimal sustainability standards, it could result in undesirable outcomes of urban inequality in terms of access to socio-economic Moreover, a balanced access to urban socio-economic  amenities is the quest of human right within the broader framework of sustainable development goals [23]. Multi-stakeholder’s involvement including urban residents in the urban changes support to achieve sustainability outcomes. Line 79

Comment 4

  • line 92-134, the information on Addis Ababa urbanization could be revisited. If Addis Ababa is representing sub-Saharan African cities, the background on sub-Saharan African urbanization has been reported, no need to be repeated for case study city.

Response 4

  • Once the Addis Ababa’s representation features were mentioned, in the following paragraphs, sub Saharan cities narration is omitted, therefore, only the case study area was discussed. Line 90-129

Comment 5

  • The research questions are not related to the background information. What is the goal of this study, what are the challenges to realize your goal?

Response 5

  • The study objective/goal, research questions, and challenges are presented as below
  • Therefore, the objective of the study is to analyze the urban land use changes and their effects on population dynamics and residents’ access to socioeconomic amenities. Research questions on: which parts of the city are accommodating land use changes, and how these urban changes are affecting the population densities and residents life qualities in terms of access to socioeconomic amenities. This study outcomes will support sustainable urban planning policies that ensures equal access to urban amenities. Line 136
  • The challenge was stated as “Lack of access to high-resolution satellite images that could have been used to determine the housing types, which were used to estimate population densities was a challenge to best meet the study objectives. However, specific locations with poor resolutions were dealt through an in-situ housing evaluations” Line 149

Comment 6

  • line 153-162 seems not to belong to the introduction section

Response 6

  • We’ve included a brief method at the end of the introduction based on the suggestions from REVIEWER 3 and the statement was below.
  • “in the final part of introduction section, explain better method and paper structure to facilitate the readers' understanding;”
  • Therefore, to give it flow with the previous paragraph, I’ve made few amendments Line 141-155

The case study area

Comment 7

  • Figure 1 legend “population density” should be the sample point for estimation of population density

Response 7

  • This is changed as per the comment

Materials and Methods

  • Methods related to remote sensing classification seems problematic

Comment 8

  • line 194-202 this section starts with summarizing and criticizing three references, without explaining how it is related to your methods.

Response 8

  • Critics like notion is omitted and statements addressed as below:
  • The urban land uses classification in this study adopts the methods applied by [40 that has relied on satellite images to manually classify urban land uses, and more specifically residential types through preset criteria of urban features. The land use classifications on the mentioned study was based on the housing features (size of the houses, number of floors and building density, quality of materials they were made out of, and other features such as streets and pavements, pools and green areas to determine the socioeconomic status of the household) [40, 41]. Such land use classification method were found to be practically useful as they were supported with an accuracy assessment through a ground trothing. Results of an accuracy assessment, the agreement/disagreement between the image based evaluated land use and the actual data is obtained by an overall accuracy value and by a kappa coefficient [40, 42]. Line 193

Comment 9

  • Line 217, “The land uses were …classification following a procedure used by [40].” Reference 40 used Landsat as input data and maximum likelihood supervised classification. If your study follows the same procedure, please clarify your input data and the classifier.

Response 9

  • In the study mentioned the urban sprawl was studied using the Landsat images, using supervised classification based on binary variables (buildup and non-buildup), but in additions to this socioeconomic segregation based on residential differentiations was done using google earth images through a preset criteria such as area occupation size, housing type, having pools or not, and this housing differentiation was not based on supervised classification of any remote sensing data. The same to this the housing types classification either to classify urban land uses and population densities estimations have applied procedures used in this study. To accommodate your comment I’ve re-stated, please check. Or the following statement was modified. “The land uses were identified for the selected years by means of visual inspection and manual classification following a procedure used by [40], which has studied socio-economic segregations based on housing differentiations.” Line 219

Comment 10

  • Since you have done land use maps for three different years, the accuracy assessment was carried out for which year and which land use classes are involved in the accuracy assessment?

Response 10

  • Groundtrothing assesement was done for the images 2020, therefore, the statement is a bit modified as follows “The land use evaluation accuracy was confirmed based on the Kappa index of agreement (KIA) [49, 50], where 10% from the total grid cells were selected from the three neighborhoods for ground trothing. The accuracy assessment has considered all random land-use classes in 2020.” Line 234

Result

  • The result section could be more informative

Comment 11

  • Line 345-352, “in 2012 the slum has been reduced to 37%...in 2020, the slum continues to reduce into 22%...” It would be better to put these two parts together, as the trend of change is the same. The same for other land use classes.

Response 11

  • As per suggestion it is stated as “According to the Figure 3, in the 2002, the Urban inner core (Aratkilo neighborhood) was dominated by slum residential houses (72%) and smaller portion of commercial buildups including hotel area (Sheraton Addis hotel). In 2012 and 2020 the slum has shown a reduction trend into 37%, and 22% respectively. On the other hand, the commercial land has been increased by 9% and 19% in 2012 and 2020 respectively. The bare land has increased into 40% in 2012 and while it slightly decreased into 36% in 2020. Unlike the previous period (2002-2012) condominium has emerged into 5% in 2020. ” Line 350

Comment 12

  • The left and right pictures in Figure 3a are not necessary. The same for Figures 3 b and c. After reduction, figure a, b and c could be combined as one.

Response 12

  • As per the comment changes are introduced into all figures 3 a,b,c,4 a,b,c,5, a,b,c.

Comment 13

  • Section 4.2 should be better structured and more compacted. The change from and to slum seems most important thus should be the focus, the description of other classes could be shortened.

Response 13

  • Under section 4.2, for the three neighborhoods descriptions are compacted as follows:
    • “In this section the land use change matrix is depicted for two successive periods. Period one covers between 2002-2012, while period two is between 2012-2020. Period one coincides with the onset of the Plan for Accelerated and Sustained Development Program (PASDEP), which comprised a comprehensive urban housing program called Integrated Housing Development Program (IHDP). Period two coincides with The Growth and Transformation Plans (GTP I & II), which has sustained the urban housing development programs. According to Figure 6, in the urban center during period one 48% of the slum remained untouched, while the remaining 52% was converted into mainly bare land (42%) and few commercial (6%) land uses. Therefore, the slum, which is dominant land use in the urban center has been the most affected land use compared to the others. Given the intention has been raising commercial land uses at the urban center, the existing commercial places remained the same at considerable extent (86%). In the same neighborhood, but in period two (2012-2020), still a considerable amount of the slum areas (57%) remained the same, except the 43% that was demolished (18%), converted into commercial area (12%), and condominiums (6%).” Line 459
    • At the urban fringe of Yeka Abado neighborhood (see Figure 7), during period one, the farmland clearance (76%) and condominium housings development (17%) were the dominant changes. After 2012, the remaining farmlands went into complete clearance and full-scale condominiums development that entails roads infrastructures. Therefore, changes in the urban fringe imply massive urban expansion has been carried out at the expense of the urban farmlands and the low-cost condominiums are the dominant buildups. Line 501
    • Altad neighborhood (relocation site for the urban core displaced community) had undergone an insignificant change during the identified periods (see Figure 8). During period one, 95% of the low-rise family houses remained unchanged, while only 4% were converted into bare land. Similarly, in the following period, the low-rising houses remained with minimal changes except some residential houses facing to the main road infrastructure converted into commercial uses and bare land in some parts 6% each. Like few commercialization of residential housings, the existing open spaces were also converted into buildups that resulted in densification in the neighborhood.  Line 524

Comment 14

  • Line 505-514, during periods one and two, if the trend of change remains the same, the description for some land use classes could be summarized, to make the text more informative

Response 14

  • This is addressed in the above bullet point or see also here
    • At the urban fringe of Yeka Abado neighborhood (see Figure 7), during period one, the farmland clearance (76%) and condominium housings development (17%) were the dominant changes. After 2012, the remaining farmlands went into complete clearance and full-scale condominiums development that entails roads infrastructures. Therefore, changes in the urban fringe imply massive urban expansion has been carried out at the expense of the urban farmlands and the low-cost condominiums are the dominant buildups. Line 501

Comment 15

  • I believe figure 6 a, b and c are not the best way to show land-use changes. These figures cannot illustrate the trend of the change, either a comparison between different land-use classes.

Response 15

  • The above your comments were already based on those figures, and here you are not recommending them. I am not clear with it. I hope your comment also should apply for the figure 7 and 8. There is also no figure 6c. But, I’ve added y axis that describes the conversions of each land uses in given period 1 and period 2. Here I wanted to describe what land uses were converted in to what land use, therefore, it will be possible to give an insight on the purposes of conversions.

Discussion

Comment 16

  • Many parts of the discussion repeat the information from the result section, whereas it should contain points of interest for discussion or reflection. It would be easier for readers if you set up the three or four points and then re-structure this section accordingly.

Response 16

  • Some duplications under this section are omitted and some contents were improved to limit the priority areas of discussion to reflect. Please refer to the discussion

Comment 17

  • Line 720 “time effective” is not convincing, as many manual works were involved.

Response 17

  • We agreed and correction is done within the statement as follows: “This study has revealed that the application of low cost and remotely workable methodologies to document urban land use dynamics, its consequences on population density changes and access to urban socio-economic amenities are possible to estimate through proxy variables.” Line 693

Comment 18

  • Line 724, “lack of high resolution satellite images such as IKONOS … for classification”, it was never mentioned what is the spatial resolution of the satellite images used in your study, so it is hard to judge whether IKONOS will improve your results.

Response 18

  • The issue of image resolution for classification is now pre informed within section 3.1 as follows. “For each of these sites historical satellite images for 2002, 2012, and 2020 were acquired and delineated on Google Earth© with 30m resolution. Next, for each selected neighborhoods a grid cells of 50m x 50m (0.25 hectare) were overlaid onto the delineated neighborhoods to evaluate the land use types.” Line 214

Reviewer 3 Report

Authors made a huge effort to improve their research and to answer to reviewers' suggestions. 

As previously said, the main topic is very interesting, well presented and discusses and perfectly fits into the Journal's scope.  Authors answer in a satisfactory way to quite all my doubts. In relation to my comment: "in paragraph 3.1, it can be helpful insert the dimensions (square kilometers or hectare) of each area" authors do not provide a useful and appropriate answer. In lines 214-217 they insert the dimension of the grid used to collect the images (and its resolution). I'd like to know the physical dimensions of each area (that are totally different,  see figure 1) in order to operate parametric assessment of the values of land use composition in the 3 different study areas. Starting from this analysis, more comments on the urban development can be add.

Author Response

Comments from Reviewer 3

Dear reviewer: Thanks again for the review and pointed out comments for the second round. Each comment is followed by a response with blue highlights. The line number or sections are also indicated for you to trace changes made in the manuscript. In addition to the comments below, grammatical and spelling corrections were

Sincerely,

General

As previously said, the main topic is very interesting, well presented and discusses and perfectly fits into the Journal's scope.  Authors answer in a satisfactory way to quite all my doubts. In relation to my comment:

Comment 1

  • "in paragraph 3.1, it can be helpful insert the dimensions (square kilometers or hectare) of each area" authors do not provide a useful and appropriate answer.
  • In lines 214-217 they insert the dimension of the grid used to collect the images (and its resolution).

Response 1

  • Image resolution and grid cells used for evaluation are mentioned in the manuscript as follows. “We mapped land use changes in the three contrasting neighborhoods that are representative for the intra-urban dynamics. For each of these sites historical satellite images for 2002, 2012, and 2020 were acquired and delineated on Google Earth© with 30m resolution. Next, for each selected neighborhoods a grid cells of 50m x 50m (0.25 hectare) were overlaid onto the delineated neighborhoods to evaluate the land use types.” Line 213

Comment 2

  • I'd like to know the physical dimensions of each area (that are totally different, see figure 1) in order to operate parametric assessment of the values of land use composition in the 3 different study areas. Starting from this analysis, more comments on the urban development can be add.

Response 2

  • According to the comment, in paragraph 3.1 the following is added:
  • “Neighborhoods that represent the city center that is slum demolition site (Aratkilo = 126.92 hectare), the middle distance from the city center, which was the first slum displaced residents relocation site, which is characterized by low-rise housings (Altad = 18.27 hectare), and the periphery, which is low cost condominium housings expansion area (Yeka Abado = 193.75 hectare) were selected.” Line 204

Round 3

Reviewer 1 Report

Concerns have been mainly addressed.

Minor comments:

  • Line 216, Please check the accuracy of this statement“… on Google Earth© with 30m resolution”, as “Google Earth includes many images collected by satellites orbiting the planet. Sourced from a variety of satellite companies…” Additionally, the spatial resolution of the images (if they are the source satellite images used in your study) shown in figure 2, is definitely higher than 30 m. For the same reason, please check the accuracy of the statement in lines 702-704.
  • Line 788-789, “through intensive, high rise residential apartments…”, This suggestion appeared without supporting arguments. Any studies to support this suggestion or any consideration on the disadvantages of using high-rise residential apartments?
  • Line 790, “ somewhere in the …” is not a scientific statement.

Author Response

Dear reviewer-

We are again grateful that much of your concerns were addressed already. The remaining comments are also dealt with as per your feedback. Under each comment, a response is provided. The line numbers, where changes are made are indicated and also those changes are in red font in the manuscript.

Again we look forward to hearing the final outcomes of revisions made.

Sincerely,

Minor comments:

Comment 1

  • Line 216, Please check the accuracy of this statement“… on Google Earth© with 30m resolution”, as “Google Earth includes many images collected by satellites orbiting the planet. Sourced from a variety of satellite companies…” Additionally, the spatial resolution of the images (if they are the source satellite images used in your study) shown in figure 2, is definitely higher than 30 m.

Response 1

  • Identification of google earth images resolution was a challenge as it holds different quality and visibility by location as well as for the different year images, therefore, the resolution is put by a range as follows.
    • “For each of these sites, historical satellite images for 2002, 2012, and 2020 were acquired and delineated on Google Earth© with 5m to 15m resolutions. These images resolution variations were observed both for different locations and different time images.” Line 216

Comment 2

  • For the same reason, please check the accuracy of the statement in lines 702-704.

Response 2

  • It is rewritten as “Also, the limitation on the urban land classification could be the lack of very high-resolution satellite images such as IKONOS, and Quickbird imageries up to 1 meter resolution, which could be the best fitting for urban dynamics analysis based on object-based urban analysis that uses both spectral and spatial information for classification. Having varied resolutions for different locations was also another limitation faced when working with google earth images [59].” Line 704

Comment 3

  • Line 788-789, “ through intensive, high rise residential apartments…”, This suggestion appeared without supporting arguments. Any studies to support this suggestion or any consideration on the disadvantages of using high-rise residential apartments?

Response 3

  • A supporting reference to same context is added as follows;
  • “Urban development programs and strategies in such sub-Saharan African cities that are running to redevelop their cities need to consider relocations with minimum negative impacts on the displaced community and even if possible redevelopment without displacement, through intensive, high rise residential apartments, which rent value could be shared and affordable through some government subsidization strategies [75].” Line 788

Comment 4

  • Line 790, “ somewhere in the …” is not a scientific statement.

Response 4

  • Changed in to “This could help urban centers to absorb higher population densities and relieve population pressure in the periphery.” Line 788-789